# NetDiff: Graph Diffusion with Improved Global Capabilities to Generate and Update Mobile Network Topologies

**Félix Marcoccia** [1 2]   **Victor Fagoo** [2]   **Gilles Monzat** [2]   **Cédric Adjih** [3]   **Thomas Watteyne** [4]   **Paul Mühlethaler** [1]

## Abstract

We introduce NetDiff, a node-conditioned denoising diffusion model that generates directional link topologies and a two-slot transmit/receive parity for mobile ad hoc networks. Directional antennas can yield high throughput but require globally consistent link decisions under sector, interference, connectivity, and half-duplex constraints. NetDiff improves global coherence with Absolute Cross-Attentive Modulation (ACAM) tokens, which provide permutation-invariant global signals and help the model match graph-level counts (e.g., density and sector usage). We also propose partial diffusion to update an existing topology with a small number of denoising steps, enabling fast reconfiguration under mobility. NetDiff reaches over 95 % of target performance with constant inference time, outperforms heuristic and omnidirectional baselines, and improves over a strong diffusion graph-transformer baseline in key metrics.

## 1. Introduction

**Problem and stakes.** We address the real-time generation of link topologies in mobile ad hoc networks equipped with directional antennas. The task is to select communication links that serve as a backbone for subsequent antenna steering, while satisfying physical constraints such as antenna sectorization, range limits, and minimized interference. A crucial constraint is that a node cannot transmit and receive simultaneously; to allow full network communication in two time slots, nodes must be assigned a transmission–reception parity, inducing a bipartite topology. Fig. 1 shows the problem we are trying to solve.

[1] Inria Paris, France [2] Thales, France [3] Inria Saclay, France [4] Independent Researcher. Correspondence to: Félix Marcoccia <felix.marcoccia@inria.fr>.

*Proceedings of the 43rd International Conference on Machine Learning*, Seoul, South Korea. PMLR 306, 2026. Copyright 2026 by the author(s).

Directional antennas can greatly increase throughput (Yi et al., 2003) but require coordinated, interdependent link decisions that are computationally expensive to optimize. Prior works address related problems through UAV placement, classical topology control, or adaptive schemes adjusting beam direction and power (Guillen-Perez & Cano, 2018; Wang et al., 2024c; Huang & Shen, 2002; Huang et al., 2002; Li & Chlamtac, 2005). Some rely on local greedy policies (Bao & Garcia-Luna-Aceves, 2002). While often sub-optimal and movement-dependent, these methods highlight that adopting directional communication is key to meeting stringent requirements and overcoming scalability limits. Combinatorial methods can find high-quality static solutions (Feng et al., 2016; Benhamiche et al., 2019) but are too slow for mobile scenarios.

**Combinatorial structure and expensive supervision.** The underlying decision problem is inherently combinatorial: links must be selected jointly under sector occupancy, interference coupling, connectivity, and parity feasibility. These constraints interact globally (e.g., a single added link may alter interference, sector feasibility, and parity validity elsewhere), making near-optimal solutions expensive to obtain even for moderate node counts. We propose to learn these expensive optimization patterns from a dataset of viable topologies, enabling fast and robust topology generation for unseen node configurations. In addition, even generating training data involves a time–accuracy trade-off: a simulation-guided quasi-exhaustive search is used when possible, with a heuristic fallback when no solution is found within a reasonable computation budget (details in App. 7).

**Why a generative model.** Inferring a set of links for a given set of nodes can in principle be formulated as a supervised learning problem. However, the inherent non-determinism of our task and the vast solution space often hinder convergence towards satisfactory solutions. In addition, generative modeling is typically better suited to capture the complex interdependencies among variables. In our case, jointly predicting both the link structure and node parity proves especially challenging for a purely supervised approach. Moreover, generative models naturally introduce

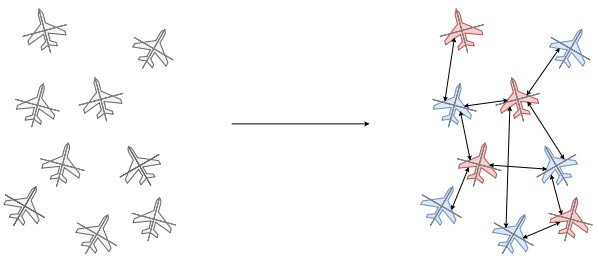

*Figure 1.* Our system consists of flying nodes equipped with sectorized antennas. We want to assign them a topology of links and a matching emission-reception parity to determine in which time slot they can emit.

a degree of smoothness and continuity in the input–output mapping, which helps to structure the solution space and facilitates the inference of new solutions by building on previously generated ones. Our generative framework can be modeled as a probabilistic problem that implies estimating the conditional joint distribution of the edges $E$ and the parity $S$ knowing $V$, which is a classical formulation for conditional generative models. Joint parity and topology generation (as opposed to a posteriori parity assignment) is particularly useful to avoid dealing with impossible direct parity assignment afterwards, which would result in expensive post-processing.

**Contributions.** We propose to extend the denoising diffusion (Ho et al., 2020) framework to solve the problem by:

- Enriching the nodes and edges with intermediate-level features, allowing to account for the emitter-receiver nature of the communications.

- Proposing additional loss terms to facilitate the compliance with the constraints.

- Introducing a novel graph-level mechanism, **(absolute) cross-attentive modulation** tokens, with a specific focus on an absolute attention variant to enhance the global coherence of the inferred edges.

- Covering the temporal evolution of the network where small and recurrent topology reconfigurations are needed because of the nodes' movement.

### 1.1. Related Work

As reviewed in (Zhu et al., 2022), five main families of graph generative models have emerged: autoregressive models, variational autoencoders, normalizing flows, generative adversarial networks, and denoising diffusion probabilistic models (DDPMs). Autoregressive models, such as Pointer Networks (Vinyals et al., 2017; Veličković et al.,

2020) or GraphRNN (You et al., 2018), generate graphs sequentially by imposing an arbitrary node ordering. This ordering bias breaks permutation invariance and becomes particularly problematic in node-conditioned frameworks, notably because autoregressive sampling becomes particularly challenging.

GAN-based architectures such as MolGAN (Cao & Kipf, 2022) or GraphGAN (Wang et al., 2017) depend on fixed-size latent noise or breadth-first sampling procedures that do not generalize to graphs with explicit node conditioning or variable size. Similarly, spectral (Martinkus et al., 2022) and flow-based methods (Shi et al., 2020) struggle to integrate continuous node features in a permutation-invariant manner, as their transformations are either global or tied to a predefined adjacency ordering. When noise is decoded through a flat MLP—as in several GAN and flow variants—conditioning on specific node features becomes ill-defined, since each noise dimension lacks a consistent correspondence to a node or edge. Variational approaches such as GVAE (Kipf & Welling, 2016) and GraphVAE (Simonovsky & Komodakis, 2018) alleviate some of these issues by learning probabilistic embeddings, yet they often assume node independence or rely on costly graph-matching during training. Additionally, Reinforcement Learning approaches (Darvariu et al., 2024; Mazyavkina et al., 2021) can address combinatorial optimization problems, but they rely on iterative procedures with non-constant runtime, often without real-time convergence guarantees, and are challenging to train when the action space is high-dimensional.

In contrast, denoising diffusion models naturally support parallel, permutation-invariant edge sampling and continuous conditioning on node features. They allow the model to learn a smooth generative process without imposing sequential dependencies, making them well suited for fast and constraint-aware topology generation. More recently, Langevin-based and diffusion models (Welling & Teh, 2011; Hoogeboom et al., 2022; Vignac et al., 2023b;a; Sun & Yang, 2023; Wang et al., 2024a) have shown strong potential for graph generation with invariance properties and adaptive noise schedules. Graph Neural Networks (GNNs), often combined with Reinforcement Learning have been widely applied in communication networks, notably for performance prediction (RouteNet (Rusek et al., 2020; Ferriol-Galmés et al., 2023)) or dynamic routing (Azzouni et al., 2017).

Though fewer works address topology generation, some methods like GCN-GAN (Lei et al., 2019) and DDPMs for wireless networks (Wang et al., 2024b) demonstrate GNNs' relevance in this context. TopoFormer (Marcoccia et al., 2025) tackles a similar problem to ours, but does not allow for joint topology-parity prediction. DDPMs are particu-

larly effective at modeling link interdependence through iterative denoising steps. This is why we seek to develop a powerful node-conditioned diffusion architecture with Net-Diff.

Various GNN architectures could be used in DDPMs for graphs: from GCNs (Kipf & Welling, 2017) and GATs (Veličković et al., 2018), to expressive GNNs (Maron et al., 2020). Recently, graph transformers (Dwivedi & Bresson, 2021) have become prominent due to their edge-aware attention mechanisms. However, capturing global structure remains challenging. To address this, DiGress (Vignac et al., 2023a) uses statistical graph properties to condition FiLM (Perez et al., 2017; Brockschmidt, 2020) layers on nodes and edges. Attention-based techniques, such as attention registers (Darcet et al., 2023) and classification tokens (Devlin et al., 2019), also help encode global information.

## 2. Solution Framework

### 2.1. Problem Statement and training data

Directional ad hoc networks must remain connected while maximizing achievable throughput under strong interference and half-duplex constraints. Optimizing the link set directly enables light, near real-time two-slot scheduling with minimal temporal coordination overhead. In our formulation, each node $v \in V$ has a parity $S_v \in \{0, 1\}$ corresponding to its transmit or receive slot. Feasible links can only connect nodes of opposite parity, which enforces a bipartite structure in the resulting communication graph. Each node is equipped with four orthogonal antennas, each covering a $\frac{\pi}{2}$ sector, and may activate at most one link per sector and slot. If multiple candidate links fall within the same sector, transmissions must be alternated, reducing throughput.

To generate training data, we rely on a reference topology-construction procedure that combines simulation-guided quasi-exhaustive search with a fallback heuristic strategy. The generated configurations reflect realistic deployments of manned aircrafts. The fleet of nodes generally follows a global direction that we use to normalize the input coordinate space; node-wise rotation invariance is not needed nor well fitted for our problem. Given a set of node positions, the primary solver explores a large space of feasible link configurations under interference, sector occupancy, and parity constraints, guided by accurate physical simulations of throughput. This process aims at identifying high-quality, constraint-compliant topologies and serves as a *reference* for learning. When no feasible or sufficiently good solution is found within a reasonable computation budget, we fall back to a constructive greedy topology-control algorithm. Detailed procedures are provided in App. 7.

### 2.2. Node-conditioned Denoising Diffusion Probabilistic Model

We then seek to find a set of edges $E = \{e_{1,1}, e_{1,2}, ..., e_{n,n}\}$ and the corresponding parity $S \in [\![0,1]\!]^n$, from the nodes information $V = \{v_1, v_2, ..., v_n\}$, which together constitute the graph $G = (V, S, E)$, with $n$ being the number of nodes. The nodes are described by their spatial coordinates $c$ and their orientation (yaw). The obtained topologies must observe the same properties as the ones produced by the data-generation reference (connectedness, antenna sectorization, interference reduction).

We use denoising diffusion to generate graphs from noisy edges in a fixed number of steps, as exemplified by Fig. 2. Its denoising schedule preserves permutation invariance, can be conditioned on the nodes' positions, allows for solutions' continuity and is expressive enough to capture the topology-parity interdependence. Node-conditioned graph denoising diffusion defines a Markovian noise process over the edges, which is used to train a denoising procedure:

$$p_\theta^t\big(\mathbf{G}^{t-1} \mid \mathbf{G}^t\big), \tag{1}$$

where $p_\theta$ is a neural network-based estimator and $t$ the diffusion step. It allows us to turn an intractable posterior approximation problem into an iterative denoising one. The noise is applied independently on each edge.

In the discrete setting (Vignac et al., 2023a), we aim to model $S$ and $E$ given $V$ by directly estimating the final edges and parity at each step:

$$\hat{p}_\theta^t\big(\mathbf{S}, \mathbf{E} \mid \mathbf{S}^t, \mathbf{E}^t, V\big) = \hat{p}_\theta^t\big(\mathbf{S}, \mathbf{E} \mid \mathbf{G}^t\big) \\ = \hat{p}_\theta^S\big(\mathbf{S} \mid \mathbf{G}^t\big) \times \hat{p}_\theta^E\big(\mathbf{E} \mid \mathbf{G}^t\big). \tag{2}$$

Its optimization is particularly simple because it is stepwise similar to a supervised 0-1 multiple classification problem where we minimize a loss that sums the reconstruction errors over both edges and parity:

$$\mathcal{L}(\theta) = \sum_{ij} l_{BCE}\big(E, \hat{p}_{(\theta,t,ij)}^E(G^t)\big) \quad + \\ \sum_i l_{BCE}\big(S, \hat{p}_{(\theta,t,i)}^S(G^t)\big). \tag{3}$$

At inference, generation of the edges and parity using the DDPM is done by progressively relying more on the model's predictions, less on the injected noise $Q$, each step $t$ follows:

$$\big(E^{t-1}, S^{t-1}\big) \sim \prod_{ij} \hat{p}_\theta^E(e_{ij} \mid G^t) \, q\big(e_{ij}^{t-1} \mid e_{ij}^t\big) \quad \times \\ \prod_i \hat{p}_\theta^S(s_i \mid G^t) \, q_s\big(s_i^{t-1} \mid s_i^t\big). \tag{4}$$

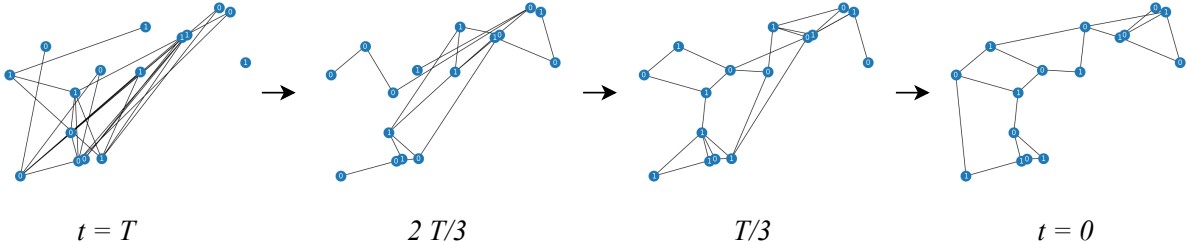

$t = T$        2 T/3        T/3        $t = 0$

*Figure 2.* Example of the generation of a 16-node topology, T = 50 steps. The topology is inferred jointly with a node parity which it should be compatible with, which softly enforces its bipartiteness.

In the discrete setting used (Vignac et al., 2023a), which we follow, noise is injected through these state transition matrices $Q$ of coefficients $q$ that determine how likely noise is to change the discrete state of the variable from one state to another. Specifically, the transition matrix at step $t$ is given by

$$\bar{Q}^t = \bar{\alpha}^t I + \bar{\beta}^t m, \text{ with } \bar{\alpha}^t = \cos^2\left(\frac{0.5\pi(t/T + s)}{1 + \epsilon}\right)$$
$$\text{and} \quad \bar{\beta}^t = 1 - \bar{\alpha}^t,$$
(5)

where $\epsilon$ is small and $m$ is one-dimensional and encodes the true distribution of possible edges. Parity noise follows $\bar{q_s}^t = \bar{\alpha}^t I + \bar{\beta}^t m_s$, with a one-dimensional $m_s$ proportional to $0.5$ for denoising. We follow this solution-space diffusion framework instead of latent diffusion, which generally implies using an encoder-based architecture that is not well suited for node-conditioned generation.

### 2.3. Network Evolution via Partial Diffusion

Physical node motion follows structured trajectories and kinematic constraints. However, it implicitly defines a bounded set of admissible perturbations in layout space. Let $X \sim \mathcal{P}_X$ denote node layouts sampled from the deployment distribution used to generate the dataset, and let $G = f(X)$ denote the corresponding topology produced by the reference solver, inducing an edge distribution $\mathcal{P}_E = f_\#(\mathcal{P}_X)$.

We approximate the admissible node-displacement set by sampling zero-mean perturbations in the normalized coordinate system, drawing random elements from this feasible neighborhood. This yields perturbed layouts $X'$, whose induced edge sets $E' = f(X')$ remain distributed according to a slightly perturbed version of $\mathcal{P}_E$. Empirically, these edge variations closely match intermediate diffusion states, which motivates partial diffusion for topology updates.

It consists in starting from the previous topology $E^{previous}$, and following a few diffusion steps, depending on how much the nodes have moved. Small movements require little or no noise, leading to only minor reconfigurations.

Larger movements are handled with more diffusion steps, proportional to a normalized measure of node displacement. In this case, the process follows the general sampling procedure but starts from the previous topology at a diffusion time proportional to the normalized displacement.

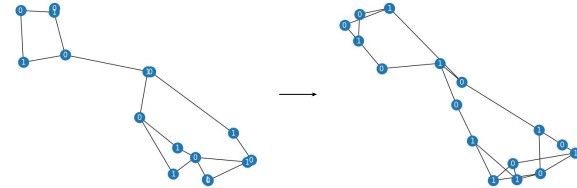

*Figure 3.* A true network topology (on the left) follows a standard reconfiguration of 15 diffusion steps. While the generated topology shares structural similarities with the base topology, important changes have been made (0.3 normalized node movement).

Fig. 3 displays the example of a network generated using the standard reconfiguration paradigm.

When nodes have only slightly moved, we use $\bar{R}^t = \bar{\alpha}^t I + \bar{\beta}^t E^{previous}$ and $\bar{R_s}^t = \bar{\alpha}^t I + \bar{\beta}^t S^{previous}$ in the noise schedule. The sampling then follows

$$\left(S^{t-1}, E^{t-1}\right) \sim \prod_{ij} \hat{p}_{(\theta,ij)}^E (e_{ij} \mid G^t) \, r(e_{ij}^{t-1} \mid e_{ij}^t) \quad \times$$
$$\prod_i \hat{p}_{(\theta,i)}^S (s_i \mid G^t) \, r_s(s_i^{t-1} \mid s_i^t).$$
(6)

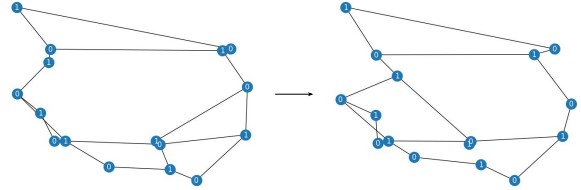

*Figure 4.* A true network topology (on the left) follows a minor reconfiguration of 10 diffusion steps. Very few links have been changed during the reconfiguration (0.1 normalized node movement).

Fig. 4 displays the example of a network generated using the minor reconfiguration paradigm.

We also follow the guided sampling diffusion framework introduced in (Dhariwal & Nichol, 2021), with $\sum_{ij} l_{BCE}(E_{ij}^{previous}, \hat{E}_{ij}^{\,t})$ replacing the classifier loss, with a scaling factor linearly depending on the amount of nodes' movement since the last reconfiguration.

In systems where node mobility is strongly influenced by their previous orientation, notably when predicting topologies at high frequency with nodes unable to perform rapid lateral movements, it is possible to design transition matrix probabilities are conditioned on the nodes' prior orientation, and train the model with the resulting noise schedule.

## 3. Solution Architecture

### 3.1. Graph Transformer with ACAM Tokens

We use a graph transformer as the denoising estimator. It takes the noisy edges $E^t$, the nodes $V$, and the diffusion timestep $t$ as inputs, and outputs a prediction of the correct edges $E$, as well as a corresponding parity $S$. The timestep $t$ is incorporated to the model using a FiLM (Perez et al., 2017) layer.

Standard attention mechanisms provide strong relational reasoning capabilities, yet they offer only limited access to *global*, *absolute* information. In graph transformers, attention is anchored to node–node interactions and is therefore inherently relational: a node can only infer properties of the whole graph indirectly, through multi-hop aggregation or repeated message passing. In practice, this makes it difficult for the model to track global properties such as density, cluster size, or the number of active sectors, especially in diffusion settings where the graph varies across steps.

A natural idea to enrich attention with global features would be to bias the queries or keys with terms that do not depend on individual node embeddings. However, injecting such absolute components separately for each node would be computationally inefficient, and the resulting information would still be entangled with per-node relational comparisons. An alternative, already explored in multiple architectures, is to introduce *virtual nodes* or *virtual tokens* (Devlin et al., 2019; Darcet et al., 2023; Hwang et al., 2022). These entities participate in message passing alongside real nodes and may serve as global aggregators. While effective, these virtual nodes inherit some of the constraints as ordinary nodes, while potentially overloading the message-passing mechanism.

To overcome these limitations, we introduce cross-attentive modulation tokens. Unlike standard virtual nodes, we *detach* the global token from the main message-passing stream. Instead of letting it attend symmetrically to all

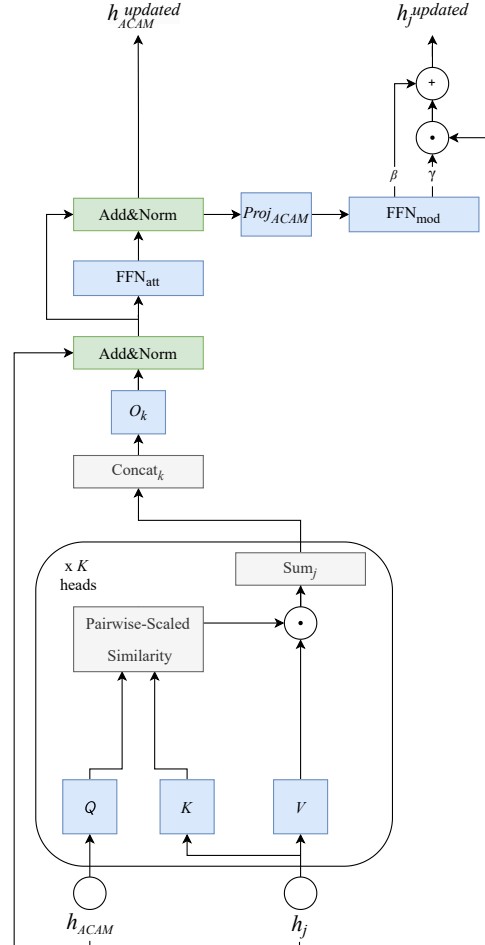

*Figure 5.* The architecture of an ACAM embedding update block.

nodes, we connect the token to the graph through a dedicated *cross-attention* mechanism. This lets the token read from the graph without interfering with node–node attention patterns. The tokens then modulate the graph through FiLM layers. While bidirectional cross-attention could be used, FiLM layers allow for a multiplicative modulation of the features which allows for expressive behavior-enforcing globally, and are more effective with when using only a few virtual nodes.

For our specific problem, where tracking global cardinalities (active sectors, total edges, local edge density, and node clusters) is crucial, we introduce **Absolute Cross-Attention (ACAM)** tokens. ACAM tokens replace the usual softmax-based attention with an *unsoftmaxed* dot-product aggregation. This yields a global descriptor that is both computationally efficient and capable of "counting". This detached and unsoftmaxed cross-attention computation can be seen as a form of *absolute attention*. Fig. 5 shows the architecture of an ACAM block.

ACAM tokens serve as global controllers whose role is to:

- Aggregate global information from the current noisy graph using absolute cross-attention.

- Inject this information back into node and edge embeddings through FiLM-based modulation.

This mechanism increases flexibility and facilitates handling difficult node layouts or unbalanced edges that would otherwise distort attention-based message passing. An ACAM token applied to nodes remains $\mathcal{O}(n)$ complex.

**ACAM Representation and Update.** At each layer $\ell$, we maintain a set of tokens $H_{\text{ACAM}}^{\ell-1} \in \mathbb{R}^{C \times d}$. We define queries, keys, and values via learnable projections $W_Q^{\text{ACAM}}, W_K^{\text{node}}, W_V^{\text{node}}$. Specifically, we compute:

$$Q_{\text{ACAM}}^{\ell} = H_{\text{ACAM}}^{\ell-1} W_Q^{\text{ACAM}}, \qquad (7)$$

$$K^{\ell} = H^{\ell} W_K^{\text{node}}, \quad V^{\ell} = H^{\ell} W_V^{\text{node}}. \qquad (8)$$

The similarity matrix is computed as:

$$A^{\ell} = \text{Sim}(Q_{\text{ACAM}}^{\ell}, K^{\ell}) \quad \in \mathbb{R}^{C \times n}. \qquad (9)$$

The function *Sim* denotes a pairwise-scaled similarity measure between queries and keys, such as a scaled dot product or a cosine similarity. In our implementation, we adopt **cosine similarity**, which provides a naturally bounded and scale-invariant measure. This choice leads to stable training dynamics and is well suited to our setting, where ACAM tokens act as latent density-aware perceivers.

For ACAM, we compute the aggregate directly without softmax:

$$Z_{\text{ACAM}}^{\ell} = A^{\ell} V^{\ell} \quad \in \mathbb{R}^{C \times d}. \qquad (10)$$

Removing the softmax allows $Z_{\text{ACAM}}^{\ell}$ to directly reflect changes in the *number* of contributing nodes or the *density* of local clusters. When multiple ACAM tokens are used, each token independently aggregates information from the graph, and their contributions are combined through a weighted summation or pooling operation. This aggregated ACAM signal forms a compact global representation of the current graph state.

The tokens are updated via a residual feed-forward block: $H_{\text{ACAM}}^{\ell} = H_{\text{ACAM}}^{\ell-1} + \phi_{\text{ACAM}}^{\ell}(Z_{\text{ACAM}}^{\ell})$.

**FiLM modulation.** To inject global information back into the graph, we pool the ACAM tokens into a global context $z^{\ell} = \text{Pool}(H_{\text{ACAM}}^{\ell})$, which is then used to condition FiLM parameters $(\gamma^{\ell}, \beta^{\ell})$ for both nodes and edges via affine projections:

$$\gamma_{\text{node}}^{\ell} = W_{\gamma,\text{node}}^{\ell} z^{\ell} + b_{\gamma,\text{node}}^{\ell}, \qquad (11)$$

$$\beta_{\text{node}}^{\ell} = W_{\beta,\text{node}}^{\ell} z^{\ell} + b_{\beta,\text{node}}^{\ell}. \qquad (12)$$

For nodes, the modulation is applied as:

$$\tilde{h}_i^{\ell} = h_i^{\ell} \odot \left(\mathbf{1}_d + \gamma_{\text{node}}^{\ell}\right) + \beta_{\text{node}}^{\ell}. \qquad (13)$$

The ACAM block is applied *after* each graph transformer layer.

The same principle applies to edges.

## 4. Additional Features and Loss Terms

We add several domain-related features and loss function terms to better solve our problem. The antenna occupation is computed following $n_i^a = \sum_j e_{ij} \cdot \delta(a(i,j) = a)$, with $\delta(a(i,j) = a)$ being the indicative function equal to 1 if $e_{ij}$ is in sector $a$, 0 otherwise, which is the number of links on a given sector of a node $i$. This feature allows each node to better count its predicted links on each sector and can help disambiguate nodes' embeddings in dense areas. During training, the penalty for a node $i$ whose the predicted edges are in an antenna sector $\widehat{a}$ is given by

$$\mathcal{L}_i^{sectors} = \sum_{\hat{a}=1}^{4} \text{ReLU}(n_{\hat{a}}^i - 1). \qquad (14)$$

We concatenate to each edge its angle with the horizontal axis. The angle attribute for a given edge $e_{ij}$ is then given by $\psi(e_{ij}) = \arctan\left(\frac{c_j^y - c_i^y}{c_j^x - c_i^x}\right)$, with $c$ being the nodes' 2D coordinates.

We apply a cosine loss, as used in (Garrido et al., 2022), in order to penalize acute angles, which are globally rare in the dataset graphs since they physically correspond to sub-optimal links. It is computed following

$$\mathcal{L}_{\cos} = 1 - \frac{1}{n} \sum_{h \in H} \frac{2}{k(k-1)} \quad \times$$
$$\sum_{1 \le i < j \le k} \hat{e}_{ij} \frac{(c_j - c_h) \cdot (c_h - c_i)}{\|c_j - c_h\| \cdot \|c_h - c_i\|}, \qquad (15)$$

$\hat{e}$ being the predicted edges and $k$ the number of neighbors of the node $h$.

Finally, to enforce the graph bipartition, we add a parity-related loss term that penalizes links between nodes of the same predicted parity:

$$\mathcal{L}_{odd} = \frac{\sum_{(i,j) \in \widehat{E}} \hat{e}_{ij} \cdot \left(1 - \left|\widehat{S}(i) - \widehat{S}(j)\right|\right)}{1 + \sum_{(i,j) \in \hat{E}} \hat{e}_{ij}},$$

$\widehat{E}$ are the predicted edges and $\widehat{S}$ the predicted parity. $\qquad (16)$

*Table 1.* Evaluation of key properties of the generated graphs compared to the target ones.

| Model | # links | Avg. link length | Link throughput | Saturation (%) ↓ |
|---|---|---|---|---|
| GraphVAE | 75.90 | 1.51 | 0.48 | 90.0±.3 |
| GT-VAE | 28.10 | 1.17 | 2.06 | 31.4±.3 |
| DDPM-GT | 23.72 | 1.05 | 3.18 | 17.6±.05 |
| DDPM-GT w/ features | 22.96 | 0.99 | 3.35 | 13.6±.05 |
| NetDiff (CAM) | 23.08 | 0.97 | 3.40 | 12.9±.05 |
| **NetDiff (ACAM)** | **22.42** | **0.94** | **3.52** | **12.1±.05** |
| Target | 21.95 | 0.85 | 3.62 | 9.7 |

*Table 2.* Avg. instantaneous throughput upper bound (Mbps).

| | Omni. | Huang et al. | MST+greedy | DDPM-GT | **NetDiff** | Target |
|---|---|---|---|---|---|---|
| **Throughput - 16 ↑** | 47.24 | 52.89 | 63.46 | 75.43 | **78.92** | 79.33 |
| **Throughput - 32 ↑** | 71.63 | 153.54 | 204.78 | 258.90 | **322.40** | 340.52 |

## 5. Results

### 5.1. Benchmarks

NetDiff uses 50 diffusion steps and 5 graph transformer blocks with features in dimension 32. Training is performed with the AdamW optimizer. The learning rate linearly decays from $10^{-3}$ to $10^{-6}$ throughout training, while the weight decay is set to $10^{-3}$ and disabled during the final 20 epochs. Models are trained with batch size 64 for at most 4700 epochs using random seed 123 for reproducibility. Early stopping based on crucial performance metrics is applied once convergence is reached. All results—excluding throughput measurements—are obtained using without any post-processing.

Experiments are conducted on an Intel Xeon(R) E5-2650v3 CPU and a Tesla T4 GPU. A full diffusion pass takes approximately 450 ms on GPU and under 2 s on CPU. The noise schedule favors redundant links, which are easier to prune than to create. Benchmarks rely on realistic, unseen 16-node graphs representative of our target networks. Although topology generation is size-agnostic, mixing different node counts during training improves generalization.

We compare NetDiff with several architectures which allow for node-conditioning and permutation invariance: Graph-VAE (Simonovsky & Komodakis, 2018), GT-VAE (same encoder with a gt-based decoder) and a discrete diffusion graph transformer baseline referred to as *DDPM-GT*. This baseline closely follows the DiGress architecture (Vignac et al., 2023a), using the same attention structure and the same statistical node and edge features. However, due to the fundamentally different application setting—directional wireless networks with geometric and scheduling constraints— several domain-specific components of DiGress are not applicable. For clarity, we there-

fore refer to this model as DDPM-GT throughout the results section. All models use the additional features and loss terms.

Table 1 shows clear differences between models; *saturation* denotes the proportion of antennas that exceed their admissible number of active links, resulting in increased interference and reduced spatial reuse. Equipping the diffusion graph transformer with domain-related features improves all metrics over the plain DDPM-GT baseline. Introducing CAM tokens further improves throughput and reduces saturation by injecting global contextual information. NetDiff with ACAM tokens provides the closest match to the target graphs, particularly in terms of interference control and saturation.

Table 2 reports the theoretical instantaneous throughput upper bound per time slot, with dynamic routing and multi-channel scheduling excluded to isolate topological effects. Diffusion-based approaches achieve substantially higher throughput than heuristic and omnidirectional baselines. NetDiff approaches the target throughput, especially on larger networks, highlighting its ability to capture long-range interference patterns.

Table 3 compares different architectural variants. CAM tokens substantially reduce node saturation compared to the DDPM-GT baseline, while ACAM tokens further improve constraint satisfaction by enabling absolute, graph-level supervision. It is also worth mentioning that, in our experiments, we observed that the advantage of CAM and ACAM tokens over the statistical features becomes more pronounced as feature dimensionality and model capacity increase, suggesting that explicit global tokens scale more favorably in expressive diffusion architectures.

*Table 3.* Constraint satisfaction (%) for different diffusion graph transformer variants.

| Model | Connected ↑ | Node Saturation ↓ | Parity ↑ |
|---|---|---|---|
| DDPM-GT | 98.50±.2 | 11.62±.05 | 98.59±.2 |
| NetDiff (CAM) | **98.68** ±**.2** | 7.98±.05 | 98.63±.2 |
| **NetDiff (ACAM)** | **98.68**±**.2** | **7.05**±**.05** | **98.64**±**.2** |

*Table 4.* Constraint satisfaction (%) using our loss function compared to standard BCE.

| Constraint | BCE | Ours |
|---|---|---|
| **Node Saturation** ↓ | 15.90±.1 | **7.05**±**.05** |
| **Parity** ↑ | 62.23±.3 | **98.64**±**.2** |

*Table 5.* Properties of the generated graphs using the two partial diffusion algorithms.

| Property | Standard (15 steps) | Minor (10 steps) |
|---|---|---|
| **# of links** | 22.79 | 22.40 |
| **Avg.link length** | 0.98 | 0.92 |
| **Link throughput** | 3.45 | 3.54 |
| **Saturation (%)** | 13.20±.1 | 12.79±.1 |
| **Continuity** | 33.22 % | 38.74 % |

Additional experiments not displayed for clarity showed that increasing the number of ACAM tokens improves the quality of the results, especially for saturation-related metrics. For example, using 16 ACAM tokens instead of 1 reduces saturation metrics by approximately **0.8%**.

Table 4 shows that the additional loss terms are essential for enforcing the targeted constraints. The cosine and sector losses reduce antenna-wise supernumerary links, while the parity loss is required to satisfy the parity constraint.

Overall, NetDiff produces topologies close to the ground truth while achieving substantially higher throughput than omnidirectional protocols. To ensure that the networks are valid and robust, we apply a lightweight post-processing routine to enforce hard constraints and restore connectivity when needed. The correction algorithm is provided in App. 10. It operates locally and preserves the intended fast inference regime.

### 5.2. Partial Diffusion for Network Evolution

We change the nodes' coordinates of true topologies following realistic trajectory patterns, and apply NetDiff on the new set of nodes, with the previous set of edges as the starting point of the diffusion. Standard reconfiguration simulates a significant change of nodes' positions while minor reconfiguration only models small changes.

*Table 6.* Constraint satisfaction (%) of the topologies obtained for standard and minor reconfigurations.

| Property | Connected ↑ | Node Saturation ↓ | Parity ↑ |
|---|---|---|---|
| Standard | 98.83 % | 7.90 % | 98.53 % |
| Minor | 98.87 % | 7.12 % | 98.66 % |

Table 5 shows that the graphs generated using partial diffusion are close to the dataset ones and that their diffusion process allows for more continuity between the previous topology and the generated one.

As shown in Table 6, the topologies obtained using our partial diffusion method present similar constraint compliance to the ones obtained using the general framework presented above.

Moreover, we observed that graphs generated using partial diffusion are about **21 %** more similar to the topology they used as a starting point than if they had been produced using the general framework. For a typical mobility scenario (where each node randomly moves with a maximal amplitude of 3/10th of the diagonal of the operational zones), we obtain the best accuracy-to-computation ratio following only 10 to 15 diffusion steps, resulting in 3.3 to 5 times faster computation.

Using the noise-less reconfiguration framework for minor reconfigurations, we produce topologies **41 %** more similar to the starting point topology, following only 7 to 10 diffusion steps.

## 6. Conclusion

NetDiff is a denoising diffusion architecture for directional network topology generation which imitates the costly optimization methods typically used to tackle such problems when real time computation is not mandatory. Absolute cross-attentive modulation tokens facilitate respecting structural constraints and allow our architecture to outperform existing diffusion graph transformers. Finally, partial diffusion enables rapid network reconfiguration under mobility by updating previous topologies with only a few denoising steps.

## Impact Statement

This paper presents work whose goal is to advance the generation and optimization of ad hoc wireless network topologies using machine learning. It designs architectural improvements that are particularly useful in contexts where the graphs to be generated must respect strong structural priors. Potential positive impacts include improved design and management of wireless communication networks. The proposed ACAM tokens and features could be effective for other graph models, and partial diffusion could also be valuable for various temporal graph settings. Potential risks include misuse in adversarial communication scenarios or over-reliance on generated topologies without sufficient validation in real deployments. We believe that these risks can be mitigated by careful evaluation, domain-specific constraints, and human oversight when deploying such methods in operational systems.

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

# Appendix

## 7. Data Generation Procedure

To generate training data, we rely on a reference topology-construction procedure combining simulation-guided quasi-exhaustive search with a fallback heuristic. Given node positions, the solver explores a large space of feasible link configurations under interference, sector occupancy, and parity constraints, guided by physical throughput simulations. It identifies high-quality, constraint-compliant topologies that serve as supervision for learning. When no satisfactory solution is found within the computation budget, we fall back to a greedy constructive topology-control algorithm.

---

**Algorithm 1** Greedy Topology Generation

---

**Require:** $V$, distance function $d(\cdot, \cdot)$, antenna sets $\{\mathcal{A}_i\}$, sector maps $\{\mathcal{Q}_i\}$, threshold $\tau$, parity $s : V \to \{0, 1, \perp\}$, throughput model $T(\cdot)$ (App. 8)

**Ensure:** $G = (V, E)$ and updated parity $s$

1: $E \leftarrow \emptyset$
2: $P \leftarrow \{\{i, j\} \subseteq V \mid (s_i \neq s_j) \text{ or } (s_i = \perp \text{ or } s_j = \perp)\}$
3: Sort $P$ by increasing $d(i, j)$
4: **while** $(V, E)$ is not connected **and** $P \neq \emptyset$ **do**
5:     Extract the shortest pair $\{i, j\}$ from $P$
6:     **if** $s_i \neq \perp$ **and** $s_j \neq \perp$ **and** $s_i = s_j$ **then**
7:         **continue**
8:     **end if**
9:     **if** $s_i \neq \perp$ **and** $s_j \neq \perp$ **and** $s_i \neq s_j$ **then**
10:         $(a^\star, b^\star, t^\star) \leftarrow \arg \max_{a \in \mathcal{A}_i, b \in \mathcal{A}_j} t(i \to j, a, b \mid E)$
11:     **else**
12:         $(a^\star, b^\star, t^\to) \leftarrow \arg \max_{a,b} t(i \to j, a, b \mid E)$
13:         $(\tilde{a}^\star, \tilde{b}^\star, t^\leftarrow) \leftarrow \arg \max_{a,b} t(j \to i, a, b \mid E)$
14:         **if** $t^\to \geq t^\leftarrow$ **then**
15:             Set $s_i \leftarrow 0$ if $s_i = \perp$; set $s_j \leftarrow 1$ if $s_j = \perp$
16:         **else**
17:             Set $s_j \leftarrow 0$ if $s_j = \perp$; set $s_i \leftarrow 1$ if $s_i = \perp$
18:             $(a^\star, b^\star) \leftarrow (\tilde{a}^\star, \tilde{b}^\star)$
19:         **end if**
20:     **end if**
21:     Let $q_i \leftarrow q_i(a^\star) \in \mathcal{Q}_i$ and $q_j \leftarrow q_j(b^\star) \in \mathcal{Q}_j$
22:     Require $\deg_{q_i}(i) \leq 0$ and $\deg_{q_j}(j) \leq 0$
23:     **if** $t(i, j) \geq \tau$ **and** sector constraints hold **then**
24:         $E \leftarrow E \cup \{(i, j)\}$
25:         Lock sectors $q_i$ and $q_j$
26:     **end if**
27: **end while**
28: **return** $(V, E, s)$

---

## 8. Sum-Rate Throughput and Interference

**Sum-rate objective.** We approximate the total network throughput by the following cumulative sum-rate proxy:

$$T = \sum_{(i,j) \in E} \mathbf{1}[s_i = 0, s_j = 1] \left(\mathcal{R}_{i \to j} + \mathcal{R}_{j \to i}\right), \quad (17)$$

$$\mathcal{R}_{i \to j} = \log_2 \left(1 + \frac{P_{i \to j}}{\sigma^2 + \mathcal{I}_{i \to j}}\right), \quad (18)$$

$$P_{i \to j} = \frac{R(i, j)}{d(i, j)^2}, \quad (19)$$

$$\mathcal{I}_{i \to j} = \sum_{\substack{(k,l) \in E \\ k \neq i, \\ s_k = 0, \, s_l = 1}} \frac{R(k, l)}{d(k, j)^2 + \epsilon} \mathbf{1}\left(\angle\left((k, l), (k, j)\right) \leq \theta_{\max}\right). \quad (20)$$

Each potential link $e_{ij} \in E$ is characterized by the following quantities:

- $d(i, j)$: the Euclidean distance between nodes $i$ and $j$;

- $R(i, j)$: the transmission power, adjusted according to $d(i, j)$ by an external power-control module;

- $\delta(i, a)$: the set of candidate links using node $i$'s antenna sector $a$ if selected (this set depends on node position and orientation).

Interference between two directed links $(i, j)$ and $(k, l)$ is modeled as a cumulative, distance-attenuated contribution:

$$\text{interference}\left((i, j), (k, l)\right) = \frac{R(k, l)}{d(k, j)^2 + \epsilon} \mathbf{1}\left(\angle\left((k, l), (k, j)\right) \leq \theta_{\max}\right), \quad (21)$$

where $\mathbf{1}(\cdot)$ equals 1 if the angular separation is below the antenna beamwidth $\theta_{\max}$ and 0 otherwise, and $\epsilon > 0$ prevents division by zero and limits near-field amplification.

Exact simulation parameters are omitted as they rely on proprietary antenna radiation patterns, propagation models, and attenuation laws specific to the deployment environment. Their physical specification is therefore out of the scope of this paper.

*Table 7.* Constraint satisfaction for 32 node-networks

| Model | Connected ↑ | Saturated ↓ | Parity ↑ |
|---|---|---|---|
| NetDiff(32) | 98.78 % | 14.08 % | 97.18 % |

## 9. Network Size Flexibility

*Table 8.* Some properties of the generated 32-node-topologies

| Property | NetDiff(32) | Target |
|---|---|---|
| **Number of links** | 54.64 | 48.22 |
| **Average link length** | 0.77 | 0.70 |
| **Link throughput** | 5.90 | 6.26 |
| **Saturated antennas** | 14.40 % | 9.91 % |

While our framework is theoretically flexible regarding the size of the input networks, obtaining similar performance on 16 or 32-node networks does require the model to "see" 32-node-networks in the training stage. We then trained the model in order to provide similar performances as the ones detailed previously for 16-node-networks while generalizing better to 32-node-networks (40+ nodes networks need to be partitioned for communication stability so the case is not covered in this work). We used a reduced dataset of 16 and 32-node networks with a 60/40 ratio to retrain our model. Since the training on various sizes could not be batched, we only retrained our model for an equivalent of 2 epochs on 15k samples.

## 10. Topology Correction after NetDiff Generation

Alg. 2 shows the correction loop used to ensure that the generated topologies and parities are valid. Its execution time depends on the quality of the generated solutions; with the settings used to obtain the results listed beforehand, the correction algorithm never runs in more than 100 ms.

---

**Algorithm 2** Topology Correction after NetDiff Generation

**Require:** Nodes $V$, raw edges $E_{\text{pred}}$, predicted parity $S_{\text{pred}} \in \{0,1\}^{|V|}$;
    sector sets $\{\mathcal{Q}_i\}$ with at most one active link per sector/slot;
    interference model $I(\cdot)$ and sum-rate model $T(\cdot)$ (App. 8);
    interference threshold $\tau_I$

**Ensure:** Corrected bipartite topology $G_{\text{corr}} = (V, E_{\text{corr}}, S_{\text{pred}})$

1:  $E \leftarrow E_{\text{pred}}$      *// initialize with raw NetDiff edges*
2:  **// 1) Sanitization: enforce hard constraints**
3:  **for** each edge $(i, j) \in E$ **do**
4:     **if** $S_{\text{pred}}[i] = S_{\text{pred}}[j]$ **then**
5:         remove $(i, j)$ from $E$ *// bipartite parity; continue*
6:         **continue**
7:     **end if**
8:     **if** SECTOROVERLOADED$(i, j, E)$ **then**
9:         remove $(i, j)$ from $E$    *// per-sector half-duplex; continue*
10:     **continue**
11:    **end if**
12:    **if** $I(i, j \mid E) > \tau_I$ **then**
13:       remove $(i, j)$ from $E$    *// cumulative interference admissibility*
14:    **end if**
15: **end for**
16: **// 2) Connectivity repair: add admissible, high-gain edges**
17: **while** NOTCONNECTED$(V, E)$ **do**
18:    $\{C_1, \ldots, C_k\} \leftarrow$ COMPONENTS$(V, E)$    *// $k > 1$*
19:    $\mathcal{P} \leftarrow \{(u, v) : u \in C_a, v \in C_b, a \neq b, S_{\text{pred}}[u] \neq S_{\text{pred}}[v]\}$
20:    filter $\mathcal{P}$ to pairs with feasible sectors and $I(u, v \mid E) \leq \tau_I$
21:    $(u^\star, v^\star) \leftarrow \arg\max_{(u,v) \in \mathcal{P}} \Delta T(u, v \mid E)$
22:    **if** $(u^\star, v^\star)$ exists **then**
23:       $E \leftarrow E \cup \{(u^\star, v^\star)\}$
24:       LOCKSECTORS$(u^\star, v^\star)$
25:    **else**
26:       **break**      *// no admissible edge: leave minor disconnectivity to scheduler*
27:    **end if**
28: **end while**
29: **// 3) Pruning: remove redundant or harmful edges**
30: **for** each edge $(i, j) \in E$ in ascending $\Delta T$-benefit order **do**
31:    **if** CONNECTED$(V, E \setminus \{(i, j)\})$ **and** $T(E \setminus \{(i, j)\}) \geq T(E) - \epsilon$ **then**
32:       $E \leftarrow E \setminus \{(i, j)\}$   *// remove low-gain redundant edge*
33:    **end if**
34: **end for**
35: **return** $(V, E, S_{\text{pred}})$

---

