# OpenReview forum: "NetDiff: Graph Diffusion with Improved Global Capabilities to Generate and Update Mobile Network Topologies"
_ICML.cc/2026/Conference — ICML 2026 regular_

### Official Review · Reviewer_9ump · 2026-03-09

**Soundness:** 2
**Presentation:** 1
**Significance:** 2
**Originality:** 2
**Overall Recommendation:** 2
**Confidence:** 4

**Summary:**

This paper studies topology generation for communication networks using graph diffusion models. The method models topology construction as a node-conditioned diffusion process that jointly predicts edges and node parity.

The paper also introduces CAM/ACAM mechanisms to capture global graph properties and proposes partial diffusion to update topologies when node positions change.

Experiments compare the method with several baselines on simulated datasets.

**Compliance With Llm Reviewing Policy:**

Affirmed.

**Final Justification:**

Some of my concerns were addressed during the rebuttal. However, as noted in my previous reply, my main concerns remain unresolved.

In particular, regarding the baselines, I do not think the statement that “DDPM-GT is essentially DiGress” is correct, since the two methods differ fundamentally in that one is continuous while the other is discrete. Overall, I do not find this justification convincing from either a technical or presentation standpoint.

I therefore maintain my score for this submission.

**Key Questions For Authors:**

1. Is the model intended to approximate the reference solver or potentially generate better topologies? If the model is trained to reproduce solutions from a reference solver and achieves similar performance, is the main benefit improved efficiency compared to running the solver directly? If not, is the goal primarily to study constrained graph generation as a machine learning problem?

2. If the model can generate better graphs than the solver, how would such improvements be evaluated?

2. Could this problem be modeled as a directed graph task instead of using node parity?

**Limitations:**

No limitations and societal impact discussions being found from my side. The largest limitations of this work remains the unclearness of the task and the marginal gain in terms of performance.

**Strengths And Weaknesses:**

*Strengths*

1. The paper addresses a meaningful application problem. Connecting graph generation with realistic communication network design is interesting, as many graph generation works focus on synthetic benchmarks.

2. The proposed framework is technically reasonable and the experiments generally support the claims that the method can generate feasible topologies under the considered constraints.

*Weaknesses*
1. The problem formulation is not clearly introduced. The task objective and the role of the reference solver are somewhat unclear (e.g., whether the goal is to approximate the solver or improve upon it). In addition, several domain-specific constraints (e.g., sector constraints, parity, interference) are introduced in a sparse way and could be explained more systematically. The motivation and mechanism of CAM/ACAM are also difficult to follow, and some evaluation metrics (e.g., link throughput, saturation) are not clearly defined. The visualization can also be more infomative by adding the node parity. The writing also remains incomplete.

2. The improvements over diffusion baselines appear relatively modest. It would be helpful to better analyze which components (features, CAM, ACAM, losses) contribute to the gains. In addition, the empirical support for ACAM is somewhat limited, and the evaluation of partial diffusion could be more thorough.

---

> ### Author Rebuttal · Authors · 2026-03-27
>
> We thank the reviewer for the detailed feedback. We agree the paper can be clearer in terms of positioning and presentation, and we address the main points below.
>
> ---
>
> **On the goal of the paper and the role of the reference solver.**
> The main point was not clearly stated enough. The topology problem we consider is **highly combinatorial** (interference, sector constraints, parity, connectivity). High-quality solutions are obtained using a **costly simulation-based quasi-exhaustive solver**, which is close to optimal but too slow for real-time or near real-time use.
>
> Our goal is not to improve this solver, but to:
> - **learn to reproduce its behavior** with a tailored diffusion model,
> - and **turn it into a fast system that can run in continuous time** using partial diffusion.
>
> This turns a heavy optimization pipeline into a neural model that runs in ** a few seconds on a modest CPU**, while producing solutions close to the ref and that can be **updated over time**.
>
> So to answer directly, the model is meant to **approximate the solver**, not outperform it. The contribution is about **efficiency and dynamic usability**. This is an **amortization approach**, where we move the cost offline and make inference cheap.
>
> ---
>
> **Can the model outperform the solver?**
> In our current setting, the model is trained to reproduce the solver, so we do not claim improvements over it.
>
> The contribution is instead:
> - taking a **near-optimal but very expensive optimization procedure**,
> - learning it with a generative model,
> - and **making it usable in practice**, especially in dynamic settings, thanks to partial diffusion.
>
> ---
>
> **On clarity of the formulation and metrics.**
> We agree that the presentation can be improved. We will:
> - clarify earlier that the task is to **approximate a constrained optimization process**,
> - present constraints (sector, parity, interference) more clearly,
> - make metrics (throughput, saturation) more explicit in the main text,
> - improve visualizations by **showing node parity (e.g., with colors)**.
>
> We also realized, partly thanks to the comments, that some content we thought was in the main paper was actually missing due to a submission issue. A duplicated placeholder table remained after an OpenReview upload problem (likely proxy-related) and replaced part of the intended content.
>
> In particular, the **additional domain-related features and losses** were not properly visible. These are important because they help the model handle counting-type effects (e.g., sector usage) and reduce overfitting. This will be fixed in the revision.
>
> ---
>
> **On CAM/ACAM and where the gains come from.**
> We agree this needs to be clearer. The paper already evaluates a progression (DDPM-GT → +features → CAM → ACAM), but we will make the contribution of each component more explicit.
>
> Empirically:
> - domain features and losses bring a large part of the gain,
> - CAM and ACAM bring additional improvements via better global consistency,
> - ACAM performs best among the global conditioning variants.
>
> We will add a clearer ablation:
>
> **Ablation results:**
>
> | Model                          | Throughput ↑ | Saturation ↓ |
> |--------------------------------|-------------|--------------|
> | DDPM-GT                        | 3.18        | 17.6         |
> | DDPM-GT + features             | 3.35        | 13.6         |
> | DDPM-GT + ACAM (no features)   | 3.39        | 13.1         |
> | NetDiff (ACAM + features)      | 3.52        | 12.1         |
>
> We also note that DDPM-GT already includes global statistics (as in DiGress), and that simpler virtual-node approaches did not work well in our setting.
>
> ---
>
> **On partial diffusion.**
> We agree this part deserves more emphasis. Partial diffusion is important because it allows:
> - **faster updates** with only a few denoising steps,
> - **continuity over time**,
> - and **much lower computation** compared to recomputing everything from scratch.
>
> This is what lets us turn a static optimization procedure into a system that can evolve over time.
>
> ---
>
> **On directed graphs vs parity.**
> The problem could be formulated as directed. However, in practice, especially with diffusion, we obtained better and more stable results by modeling **parity at the node level**.
>
> This directly enforces the half-duplex constraint and ensures bipartiteness. In contrast, a directed formulation requires the model to implicitly learn scheduling consistency, which we found less stable.
>
> ---
>
> **In short.**
> The main idea is simple:
> - the problem is **complex and combinatorial**,
> - good solutions come from a **heavy, near-optimal solver**,
> - we **learn to imitate this solver**,
> - and we **make it usable in practice** with fast inference and smooth updates over time (partial diffusion).
>
> We agree this was not stated clearly enough and will improve it in the revision.

---

> > ### Author Rebuttal · Reviewer_9ump · 2026-04-02
> >
> > Thank you for the rebuttal. I appreciate the effort, and I agree that the response helps make the intended positioning of the paper clearer. However, I still see my concerns as only partially resolved. Discussed as follows:
> >
> > 1. **The experiments still feel quite preliminary.** First, the comparisons are mostly against few basic and relatively old baselines (e.g., GraphVAE from 2018 and the DDPM from 2020), and the diffusion comparison is fairly narrow. I also did not see comparisons with stronger or more recent graph generation baselines, although the paper does discuss works such as DiGress in the related work. In addition, I think DDPM-GT + features is the more direct baseline here rather than the plain DDPM-GT, since adding domain-specific features is already a very standard way to strengthen topology-generation models. From that perspective, the gap between DDPM-GT + features and the final model does not look very evident.
> >
> > 2. **The clarity issue is still not fully addressed.** I appreciate the clarification that part of the presentation issue may be related to a submission/upload problem. At the same time, without a revised PDF, it is hard to tell how much this would improve the paper in practice. In its current form, I still found parts of the paper difficult to follow, and the presentation/formatting also feels somewhat incomplete in places. So while this may well be improvable in a next version, based on the present submission and rebuttal, I still view the concern as only partially resolved.

---

> > > ### Author Response · Authors · 2026-04-02
> > >
> > > Thank you for the fast rebuttal comment:
> > >
> > > **On the experiments and baselines**  We absolutely agree on the fact that the most relevant baseline is the DDPM-GT with features. It was already included in the most important table, and will also be used (instead of the DDPM-GT). Importantly, this DDPM-GT is essentially **DiGress** but we chose a more generic name because some of the features of DiGress are specific to its application domain. DiGress is a ICLR 2023 paper and is, to our knowledge and experiments, the best graph architecture for node-conditioned generation. As discussed in the paper, autoregressive works are particularly challenging to use as they generally require sequential sampling (extremely tedious with the node-positioning condition) and latent diffusion, which is a growing subject, is not straightforward to use in such node-conditioned setting (adapting it did not show any improvement in our early experiments, the latent vector concatenated/added/FiLMed to the nodes' positions was constantly ignored). Roto-invariance is not suited to our setting, so E(3) works are not particularly suited either.
> > >
> > > NetDiff constantly displays noticeably improved results over the baseline, significantly reducing the gap between the generated topologies and the target, but we agree that the improvement can be seen as incremental. Bringing a noticeable improvement, using ACAM tokens, as well as a solution continuity over time using partial diffusion to a modern, highly capable architecture to better tailor it to our task (low number of nodes, high solution heterogeneity, both local constraints and structural patterns) is the aim of our work.
> > >
> > > **On clarity** We understand the concern, the updated version was indeed flawed in terms of clarity and presentation. The current version of the paper is largely improved, making it a cleared and overall better read. Any further comments will allow us to improve it even further. We included explanations and inductive links where we understood that the paper did not make the points clear enough, the global problem and the aim of our model has been rewritten in more explicit terms. We improved the overall layout and the result tables. We emphasized the most important points, notably regarding experiments.

---

### Official Review · Reviewer_ue7E · 2026-03-11

**Soundness:** 3
**Presentation:** 3
**Significance:** 3
**Originality:** 2
**Overall Recommendation:** 4
**Confidence:** 5

**Summary:**

This manuscript presents an interesting node-conditioned diffusion model named NetDiff for the generation of communication topologies in directional mobile ad hoc networks. The model is capable of generating the link structures and the transmission/reception parity subject to antenna sectors, interference, connectivity, and half-duplex communication. For better global coherence, the authors have proposed ACAM tokens, which capture global information from the graph and inject this information into the node and edge embeddings using FiLM modulation. Moreover, the authors have proposed partial diffusion for efficient updates of existing topologies with a minimal number of denoising steps for supporting dynamic network reconfiguration.

**Compliance With Llm Reviewing Policy:**

Affirmed.

**Final Justification:**

The authors' arguments successfully counter my criticisms of scalability, the importance of the reference solver, the contribution made by the ACAM tokens, and the experimental setting. The additional experiments on larger networks (64 nodes) and the explanation of how the framework is an amortized approximation to an expensive combinatorial problem make the work more practically valuable.

While the overall framework draws from the literature on graph generation via diffusion and architectural contributions are not very significant. The proposed modifications like ACAM for increased coherence and partial diffusion for topology updates are justified and practically valuable within the scope of the considered applications. Despite the mentioned limitations and open problems, such as simulations, small size of the datasets, and absence of real-world applications, I believe that this paper makes a significant contribution towards solving constrained network topology design through generative modeling. Therefore, I would like to maintain my positive score.

**Key Questions For Authors:**

1. Can the authors provide some information about ablation studies on the effect of ACAM tokens in comparison to other architectural components and other loss terms?

2. Majority of the experiments seem to be run on 16-node networks. Is it because the authors are limited by the computational power required for the model and the simulation? How does the model perform on larger networks (e.g., 64 nodes or 128 nodes), and what are the computational limitations in such cases?

3. How much of the performance gain is due to the model and how much is due to the post-processing step in correcting the generated topologies? A brief explanation would suffice.

4. Since the training data is generated by a reference solver under specific assumptions, how well does the model generalize to changes in the model interference types, node distributions, or antenna configurations?

**Limitations:**

Yes

**Strengths And Weaknesses:**

Strengths
1. The problem addressed is topology generation for directional mobile ad-hoc networks. This is an interesting combinatorial network optimization problem that involves interference constraints, sector constraints, connectivity constraints, and scheduling constraints.

2. The model generates topology as well as transmission-reception parity. This avoids the need for complex post-hoc parity generation.

3. ACAM tokens are introduced to the model to facilitate the incorporation of global information into the graph transformer via cross-attention and FiLM modulation. This addresses the difficulty in capturing global information in the graph diffusion model.

4. The model introduces an intriguing partial diffusion to facilitate topology updates from a given network topology which minimizes the number of denoising steps required for network updates.

5. The NetDiff model was evaluated against various generative models such as GraphVAE, GT-VAE, graph transformer, and the diffusion graph transformer which proved its excellence in providing better throughput as well as minimizing antenna saturation.

Weaknesses
1.  The generative model mostly builds upon existing node-conditioned diffusion models for graphs, and the training objective is similar to that of traditional discrete diffusion models. The main architectural novelty is ACAM tokens but the overall framework is otherwise close to prior diffusion-based graph generators.

2. Training data are artificially generated using a simulation based quasi-exhaustive solver. This implies the model effectively learns to imitate solutions of an expensive optimization process.

3. Most evaluations are performed on realtively small 16-node topologies, with only a few examples on larger graphs. It is unclear how well the approach will scale to larger or more realistic network sizes.

4. The throughput measure uses a simplified interference model with some parameters kept secret, makes it harder to fully assess how the generated topologies would perform under real time network settings.

---

> ### Author Rebuttal · Authors · 2026-03-27
>
> We thank the reviewer for the careful reading and helpful comments. We agree the paper can be clearer about what comes from prior diffusion models and what is specific to our contribution, and we will improve this in the revision.
>
> ---
>
> **On novelty.**
> We agree that the overall framework builds on prior discrete graph diffusion models (e.g., DiGress). Our contribution is not a new diffusion paradigm, but a focused extension to a challenging combinatorial topology problem:
> (i) **ACAM tokens** to improve graph-level coherence (e.g., counting, density, sector usage),
> (ii) **partial diffusion** to efficiently update topologies over time.
>
> We want to stress that the gains do not come from a single component. We evaluate a clear progression (DDPM-GT → +features → CAM/ACAM), which lets us better understand where improvements come from. Empirically, the gains are roughly split between:
> - domain-specific features and losses (a large part of the improvement),
> - CAM/ACAM (additional gain through better global consistency),
>
> with ACAM being the strongest variant.
>
> We will make this clearer by adding an explicit ablation (ACAM without features):
>
> **Ablation results:**
>
> | Model                          | Throughput ↑ | Saturation ↓ |
> |--------------------------------|-------------|--------------|
> | DDPM-GT                        | 3.18        | 17.6         |
> | DDPM-GT + features             | 3.35        | 13.6         |
> | DDPM-GT + ACAM                 | 3.39        | 13.1         |
> | NetDiff (ACAM + features)      | 3.52        | 12.1         |
>
> We also note that DDPM-GT already includes global statistics (as in DiGress), and that simpler virtual-node or register approaches did not work well in our setting.
>
> Additionally, **post-processing** is only used for throughput computation, as this metric cannot be evaluated on physically invalid topologies. Importantly, all other metrics are computed on raw model outputs without any post-processing, so the improvements cannot be attributed to external heuristics.
>
> ---
>
> **On training data from an expensive solver.**
> We agree with the reviewer’s observation. The model is indeed trained to imitate a costly optimization process. This is intentional: the problem is highly combinatorial and too expensive to solve online. The idea is to *amortize* this cost:
> - expensive simulation-based optimization offline,
> - fast inference at test time.
>
> This is also where **partial diffusion** matters: instead of recomputing everything from scratch, we can update an existing topology with only a few denoising steps, which makes dynamic updates much more efficient.
>
> ---
>
> **On scalability.**
> While the main paper focuses on 16-node graphs, we also ran preliminary larger-scale experiments. Results on 64-node graphs show that the method still works, with some expected degradation (these are quick experiments, not yet fully optimized):
>
> **64-node results (preliminary):**
>
> | Setting            | Throughput ↑ | Connected ↑ | Parity ↑ | Saturation ↓ |
> |--------------------|-------------|-------------|----------|--------------|
> | 64-node NetDiff    | about 850         | 97.9        | 96.2     | 17.3         |
>
> The model also generalizes well to unusual node layouts, as the dataset includes many edge cases. That said, the antenna configuration is fixed, so significantly different physical setups would require retraining. Also, very large monolithic graphs are not our main target, as real systems would typically be partitioned.
>
> ---
>
> **On throughput and physical details.**
> Throughput is computed using the explicit sum-rate/interference formulation given in the appendix, which is theoretically grounded. The only part we cannot disclose is the internal simulator used to generate training data (due to industrial constraints). We will make this distinction clearer. We also provide a synthetic notebook so the method can be tested in a fully transparent setting.
>
> ---
>
> **On missing content.**
> We realized that the **additional features** section was not properly visible in the main paper due to a duplicated placeholder table left during submission (likely due to a proxy-related upload issue). These features are important because they help the model handle counting-related effects (e.g., sector usage) and reduce overfitting. This will be fixed in the revision.
>
> ---
>
> **In short.**
> The contribution is to take a **complex and costly combinatorial optimization process** and turn it into a **fast neural model** that can run in a few seconds on CPU, while producing high-quality solutions that can be efficiently updated over time with partial diffusion.

---

> > ### Author Rebuttal · Reviewer_ue7E · 2026-04-03
> >
> > I appreciate the authors for conducting additional tests for 64-node settings to test scalability of the model. Thank you addressing my questions and view points.

---

### Decision · Program_Chairs · 2026-04-30

**Decision:**

Accept (regular)

**Comment:**

The paper introduces NetDiff, a node-conditioned denoising diffusion model for generating directional link topologies and a two-slot transmit/receive parity, a combinatorial optimization problem with excessive complexity. NetDiff improves global coherence with Absolute Cross-Attentive Modulation (ACAM) tokens, which provide permutation-invariant global signals and help the model match graph-level counts. It also proposes partial diffusion to update an existing topology with a small number of denoising steps, enabling fast reconfiguration allowing for applications with fast dynamics. The solution approximates the solution over 95 % of target performance and improves over a strong diffusion graph-transformer baseline on key metrics. While the propose algorithm cannot possibly defeat all thinkable solutions, they have shown the value of their contributions through careful experiments and computation.

The paper is interesting and well written. The proposed methods exploits Token selection and partial diffusion to significantly improve on the best known algorithm in the literature. I have read the paper carefully against the comments of both reviewers and to me the authors response address issues raised. On the main remaining item for the second reviewer, in terms of benchmarks for comparison, I find the authors' response way more convincing that the reviewer.